# Quantum non-demolition readout of an electron spin in silicon

J. Yoneda [1,2 ✉], K. Takeda [1], A. Noiri [1], T. Nakajima [1], S. Li[1], J. Kamioka[3], T. Kodera [3] & S. Tarucha[1 ✉]

While single-shot detection of silicon spin qubits is now a laboratory routine, the need for quantum error correction in a large-scale quantum computing device demands a quantum non-demolition (QND) implementation. Unlike conventional counterparts, the QND spin readout imposes minimal disturbance to the probed spin polarization and can therefore be repeated to extinguish measurement errors. Here, we show that an electron spin qubit in silicon can be measured in a highly non-demolition manner by probing another electron spin in a neighboring dot Ising-coupled to the qubit spin. The high non-demolition fidelity (99% on average) enables over 20 readout repetitions of a single spin state, yielding an overall average measurement fidelity of up to 95% within 1.2 ms. We further demonstrate that our repetitive QND readout protocol can realize heralded high-fidelity (>99.6%) ground-state preparation. Our QND-based measurement and preparation, mediated by a second qubit of the same kind, will allow for a wide class of quantum information protocols with electron spins in silicon without compromising the architectural homogeneity.

[1] RIKEN Center for Emergent Matter Science, RIKEN, Saitama 351-0198, Japan. [2] Center for Quantum Computation and Communication Technology, School of Electrical Engineering and Telecommunications, The University of New South Wales, Sydney, NSW 2052, Australia. [3] Department of Electrical and Electronic Engineering, Tokyo Institute of Technology, Tokyo 152-8550, Japan. ✉email: jun.yoneda@alum.riken.jp; tarucha@riken.jp

Single electron spins confined in silicon quantum dots hold great promise as a quantum computing architecture with demonstrations of long coherence times[1], high-fidelity quantum logic gates[2–4], basic quantum algorithms[5], and device scalability[6]. However, the ability to measure a qubit in a single-shot QND manner has been lacking, despite its pivotal role in quantum error correction and quantum information processing, as well as its centrality to quantum science[7–9]. An ideal single-shot QND readout process would, in addition to yielding an eigenvalue of the observable with projection probability for the input state (measurement), leave the system in the projected input state (non-demolition), meaning that the measurement is repeatable and that a posterior state can be predicted based on the eigenvalue obtained (preparation)[7]. These features contrast with conventional readout schemes of a silicon spin qubit, which inherently demolish the spin state by mapping it to a more readily detectable, charge degree of freedom[1–6,10]. Such spin-to-charge conversion techniques are employed to facilitate to measure the small magnetic moment of a single electron spin within its relaxation time, which, although exceptionally long for a solid-state quantum system, is limited to the millisecond timescale. A QND readout requires a mechanism to exquisitely expose the system to external circuitry for readout while maintaining the coherence and integrity of the qubit. Synthesizing an ancilla system which can be repeatedly initialized, controlled conditionally on the qubit state and separately measured, all on the microsecond timescale, constitutes a major challenge for the QND readout of a silicon electron spin qubit.

In this work we demonstrate repeatable measurements of a silicon electron spin qubit. We use a neighboring electron spin as an ancilla, with which we can perform a QND qubit readout at a 60 μs repetition cycle through a conditional rotation and spin-selective tunneling. The highly QND nature is evidenced by the strong correlation between successive ancilla measurement outcomes. We take advantage of the repeatability and construct a QND qubit readout from $n$ consecutive ancilla measurements to improve the overall performance. For complete characterization as a QND readout process, we identify and evaluate three key metrics[7]: the non-demolition fidelity ($F_{QND} = 99\%$ for $n = 1$); the measurement fidelity ($F_M = 95\%$ for $n = 20$); the preparation fidelity ($F_P = 92\%$ for $n = 20$). (The numbers are the average of the spin-down and -up cases.) The non-demolition and preparation fidelities ($F_{QND}$ and $F_P$) which are dissimilar to those in the destructive readout illustrate the distinct properties of the QND readout. We further show that the repetitive readout scheme allows us to preselect the cases where the qubit state is prepared with fidelities >99.6%.

## Results

**Ising-coupled qubit-ancilla system.** Our qubit and ancilla are electron spins confined in a double Si/SiGe quantum dot (Fig. 1a) with natural isotopic abundance[11]. Spin states can be discriminated and reinitialized within 30 μs relying on energy-selective spin-to-charge conversion[10,12] and the reflectometry response from a neighboring charge sensor (see Methods for details). An on-chip micromagnet magnetized in an external magnetic field $B_{ext} = 0.51$ T separates the resonance frequencies of the qubit and ancilla spins by 640 MHz (centered around ~16.3 GHz). This enables frequency-selective electric-dipole-spin resonance rotations of individual spins at several MHz and ensures that the exchange interaction of ~MHz is well represented by the Ising type with minimal disturbance to the spin polarizations[13,14].

We correlate the ancilla and the qubit spins by a controlled-rotation gate (Fig. 1b). During a square gate-voltage pulse for a duration $t_{CZ}$ at a symmetric operation point, the ancilla spin acquires a qubit-state-dependent phase due to enhanced exchange coupling[3,15]. A Hahn echo sequence converts this phase to the ancilla spin polarization, in a robust manner against a slow drift of the ancilla precession frequency and the qubit-state-independent phase induced by the square gate-voltage pulse (~20π per μs) and the microwave bursts (~0.16π)[16,17]. We extract the qubit-dependent phase shift by changing the prepared qubit state by the microwave burst time $t_b$ (Fig. 1c). The extracted phase grows linearly with $t_{CZ}$ (Fig. 1d), consistent with an induced excess exchange coupling $J$ of 0.94 MHz. Choosing $t_{CZ} = 0.53$ μs ($= 1/2J$) and an appropriate projection phase $\theta$, we can implement a conditional rotation which maps the qubit state to the ancilla spin, allowing for the ancilla-based measurement of the qubit spin.

**Demonstration of repetitive readout.** We now demonstrate that the ancilla can be repeatedly entangled with the qubit and measured, using a sequence shown in Fig. 2a. After preparing the qubit state by microwave control, we repeat 30 cycles of a controlled-rotation gate and the ancilla measurement and reinitialization, until we destructively read out and reinitialize the qubit. We use $m_i$ and $q$ to denote the outcomes of the $i$-th ancilla measurement (with $i = 1, 2, \dots 30$) and the final qubit readout, respectively. Remarkably, all ancilla measurement outcomes show clear Rabi oscillations (Fig. 2b), indicating each functions as a single-shot QND readout of the qubit. Strong correlations between successive measurements, a hallmark of the QND readout, are verified from joint probabilities $P(m_1 m_2)$, see Fig. 2c.

The Rabi oscillation visibility of $m_i$ is affected by both the probability distribution $p_{i-1}^{\downarrow(\uparrow)}$ of the prepared qubit spin state $s_{i-1}$ and the $i$-th QND measurement fidelity $f_i^{\downarrow(\uparrow)}$ given $s_{i-1} = \downarrow (\uparrow)$. We separate the error in the prepared qubit spin state (during the process of initialization, rotation, and preceding ancilla measurements) from the measurement infidelity[18] by expressing the joint probability $P(m_i m_{30})$ as

$$P(m_i m_{30}) = \sum_{s = \downarrow, \uparrow} p_{i-1}^s \Theta_{s, m_i}(f_i^s) \Theta_{s, m_{30}}(g_i^s). \tag{1}$$

Here $g_i^{\downarrow(\uparrow)}$ denotes the measurement fidelity of $m_{30}$ for $s_{i-1}$ prepared in $\downarrow(\uparrow)$, and $\Theta_{s,m}(f)$ equals $f$ when $s = m$ and $1 - f$ when $s \neq m$. We model $p_{i-1}^{\downarrow(\uparrow)}$ by an exponentially decaying Rabi oscillation[11] and obtain $p_i^{\downarrow(\uparrow)}, f_i^{\downarrow(\uparrow)}$, and $g_i^{\downarrow(\uparrow)}$ as a function of $i$ (see Methods and Supplementary Fig. 2). We find that $f_i^{\downarrow(\uparrow)}$ is essentially $i$-independent as expected, with the average 85% (75%) for $i = 1$–20.

**Characterization of the QND readout.** A distinct feature of the QND readout is that it is repeatable, meaning we can potentially gain more accurate information about the qubit state from consecutive measurements. In the following, we leverage this potential by constructing a cumulative QND readout from $n$ outcomes, $\mathbf{m}_n = \{m_1, m_2, \dots m_n\}$ which yields estimators $\sigma$ for $s_0$ (the input qubit state, projected to either spin-down or -up) and $\varsigma$ for $s_n$ (the posterior qubit state), see Fig. 3a. We characterize its performance as a QND readout as a function of $n$, through three key fidelity figures of merit, $F_{QND}$, $F_M$, and $F_P$. These fidelities are, as depicted in Fig. 3a, defined by the correspondences between the estimators ($\sigma$ and $\varsigma$) and/or the qubit states before and after the process ($s_0$ and $s_n$). Importantly, these together will enable us to test all key criteria that the QND readout should satisfy[7]—i.e., non-demolition ($F_{QND}$), measurement ($F_M$), and preparation ($F_P$).

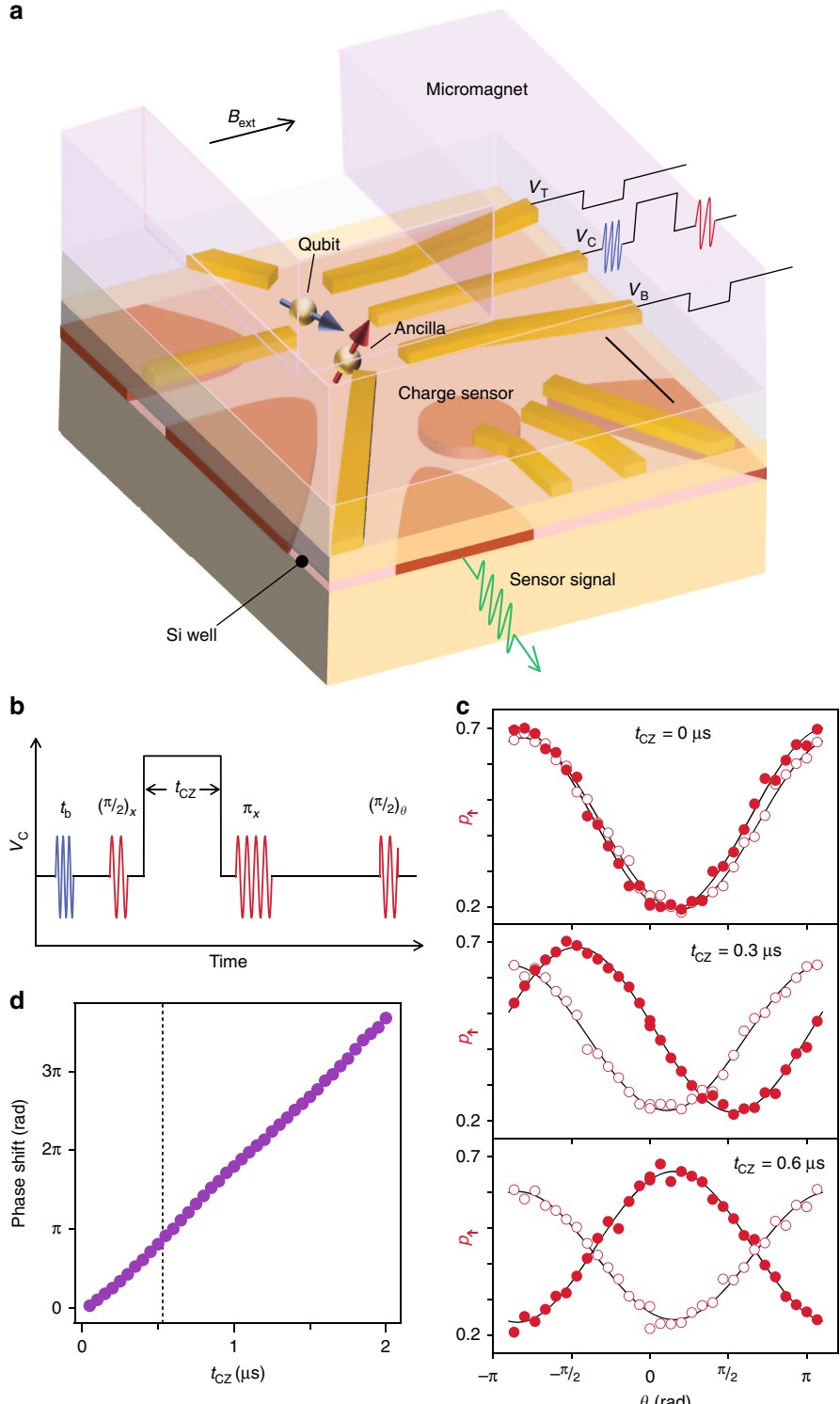

**Fig. 1 Qubit and ancilla system. a** Schematic of a device. The qubit spin (blue) and the ancilla spin (red) are hosted in two singly-occupied dots in a silicon quantum well layer. A proximal single electron transistor serves as a charge sensor. Scale bar: 200 nm. **b** Control pulse. Two microwave tones (represented by different colors) are used to selectively rotate qubit and ancilla spins. A controlled-phase shift is induced by applying square pulses simultaneously to $V_T$, $V_B$ as well as to $V_C$. **c** Ancilla spin-up probability after an entangling gate pulse. Traces with and without a π pulse applied to the qubit spin are plotted with filled and open symbols, respectively. **d** Measured controlled-phase accumulation. The dotted line indicates $t_{CZ} = 0.53$ μs used for a conditional rotation.

We first assess the non-demolition fidelity $F_{QND}^{\downarrow(\uparrow)}$, which addresses the requirement that the measured observable (spin-down or up) should not be disturbed. It represents the correlation between the projected input ($s_0$) and posterior ($s_n$) qubit states, and unlike the other two fidelities, it is expected to decrease as $n$ is increased. $F_{QND}^{\downarrow(\uparrow)}$

can be defined using the conditional probability of $s_n$ given $s_0$ as $F_{QND}^{\downarrow(\uparrow)} = P(s_n = s_0|s_0 = \downarrow (\uparrow))$. It follows from this definition that $p_n^\downarrow = F_{QND}^\downarrow p_0^\downarrow + (1 - F_{QND}^\uparrow)p_0^\uparrow$. The results obtained from the fit to this equation is shown in Fig. 3b, where $F_{QND}^{\downarrow(\uparrow)}$ gradually

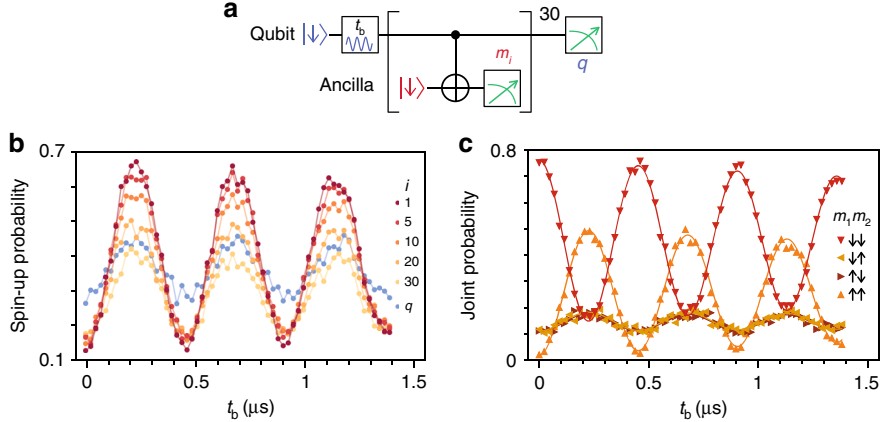

**Fig. 2 Repetitive readout. a** Quantum circuit for repetitive measurements. **b** Spin-up probabilities of the $i$-th ancilla measurement (only $i = 1, 5, 10, 20,$ and 30 are shown for brevity) and the final qubit readout ($q$) out of 1000 events. Note the oscillation visibility for $q$ is influenced by the compromised sensor sensitivity. **c** Probabilities of the four joint outcomes for the first and second ancilla measurements. The triangle symbols represent the experimental data, and the solid lines are the fit results to the model which takes into account preparation and measurement imperfections.

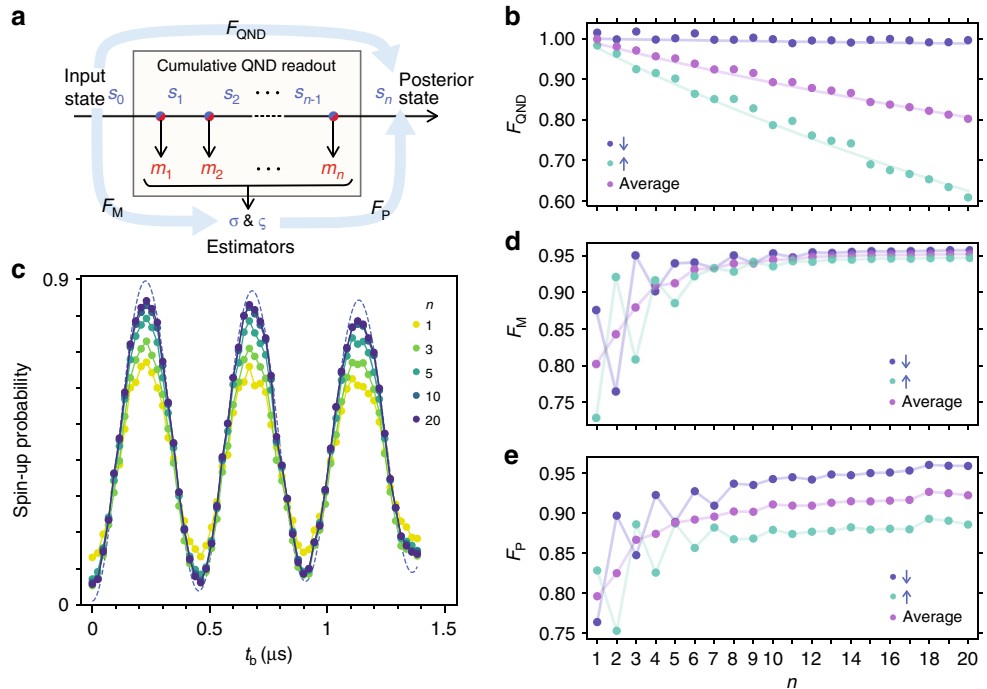

**Fig. 3 Cumulative QND readout and fidelities. a** Diagram for the cumulative readout protocol and fidelity definitions. We regard $n$ consecutive ancilla measurements (with outcomes $m_1, m_2... m_n$) as a single QND readout (with estimators $\sigma$ and $\varsigma$). The projected input spin state $s_0$ (either ↓ or ↑) changes to the posterior state $s_n$ after the process. The ideal QND measurement would give $s_n$ identical to $s_0$ (non-demolition); $\sigma$, identical to $s_0$ (measurement); and $s_n$, identical to $\varsigma$ (preparation). $F_{QND}$, $F_M$, and $F_P$ quantify these properties. **b** $F_{QND}^{\downarrow}$, $F_{QND}^{\uparrow}$, and $(F_{QND}^{\downarrow} + F_{QND}^{\uparrow})/2$ after $n$ repetitive measurements. The solid lines show the values expected from the extracted $T_1^{\downarrow(\uparrow)}$. **c** Rabi oscillations of the qubit spin acquired from multiple ancilla measurements. Plotted with a dashed curve is the estimated true qubit spin-up probability, $p_0^{\uparrow}$, consistent with a Rabi oscillation at a 630 kHz frequency detuning. **d** $F_M^{\downarrow}$, $F_M^{\uparrow}$, and $F_M^{avg} = (F_M^{\downarrow} + F_M^{\uparrow})/2$ as a function of $n$. State-dependent single-shot measurement fidelities ($f_i^{\downarrow} > f_i^{\uparrow}$) produce even-odd effects of $F_M^{\downarrow(\uparrow)}$, whereas the average $F_M^{avg}$ increases monotonically. Note that $F_M^{avg}$ can be related to the measurement visibility[12] $V$ through $V = 2F_M^{avg} - 1$. **e** $F_P^{\downarrow}$, $F_P^{\uparrow}$, and $(F_P^{\downarrow} + F_P^{\uparrow})/2$ as a function of $n$.

decreases to 99% (61%) as $n$ is increased up to 20. By modeling the $n$ dependence of $p_n^{\downarrow}$ (see Methods), we estimate $F_{QND}^{\downarrow(\uparrow)}$ for $n = 1$ to be 99.92% (97.7%), corresponding to the longitudinal spin relaxation time $T_1^{\downarrow(\uparrow)}$ of 78 ms (2.5 ms) given the 60 μs cycle time.

The second requirement for the QND readout is that the measurement result should be correlated with the input state following the Born rule. We test this through the measurement

fidelity defined as $F_M^{\downarrow(\uparrow)} = P(\sigma = s_0 | s_0 = \downarrow (\uparrow))$, where $\sigma$ is the estimator for the input qubit state $s_0$ based on measurement results $\boldsymbol{m}_n$. When $\sigma$ is the more likely value of $s_0$, $P(\boldsymbol{m}_n | s_0 = \sigma) > P(\boldsymbol{m}_n | s_0 = \bar{\sigma})$ with $\bar{\sigma}$ denoting the spin opposite to $\sigma$. We calculate these likelihoods using a Bayes model that assumes spin-flipping events (see Methods). $\sigma$ shows larger Rabi oscillations as $n$ is increased (Fig. 3c), demonstrating

$F_M^{\downarrow(\uparrow)}$ enhancement by repeating ancilla measurements in our protocol. We obtain $F_M^{\downarrow(\uparrow)}$ (Fig. 3d) through $P(\sigma = \downarrow) = F_M^\downarrow p_0^\downarrow + (1 - F_M^\uparrow) p_0^\uparrow$. While $F_M^{\downarrow(\uparrow)} = 88\%$ (73%) for $n = 1$, it reaches 95.6% (94.6%) for $n = 20$, well above the measurement fidelity threshold for the surface code[8].

The last feature of the QND readout to be evaluated is the capability as a state preparation device. In order to quantify how precisely our cumulative QND readout process prepares a definite qubit state, we define the preparation fidelity $F_P$ as the conditional probability of $s_n = \varsigma$ given the estimator $\varsigma$ for the posterior qubit state $s_n$, i.e., $F_P^{\downarrow(\uparrow)} = P(s_n = \varsigma | \varsigma = \downarrow (\uparrow))$. We emulate the most relevant situation of a completely unknown input[7] by using data with $0.08\,\mu s < t_b < 1.3\,\mu s$, for which $p_0^\downarrow = 0.500$. To optimally determine $\varsigma$ from $\boldsymbol{m}_n$, we again apply the Bayes' rule (Methods) and compare the likelihoods $P(\boldsymbol{m}_n | s_n = \downarrow)$ and $P(\boldsymbol{m}_n | s_n = \uparrow)$. We estimate $s_n$ from another estimator $\sigma'$ and convert the conditional probability $P(\sigma' = \varsigma | \varsigma = \downarrow (\uparrow))$ to $F_P^{\downarrow(\uparrow)}$ using the measurement fidelity of $\sigma'$ for $s_n$ (Methods). We obtain $F_P^{\downarrow(\uparrow)} = 76\%$ (83%) for $n = 1$, which increments to 95.9% (88.6%) for $n = 20$ (Fig. 3e).

**Heralded high-fidelity state preparation.** It is worth noting that for $n \geq 2$, these likelihoods $P(\boldsymbol{m}_n | s_n = \varsigma)$ can signal events where we have higher confidence in the final spin state. To explore this potential of heralded high-fidelity state preparation, we calculate the likelihood ratio $\Lambda^\varsigma = P(\boldsymbol{m}_{10} | s_{10} = \varsigma) / P(\boldsymbol{m}_{10} | s_{10} = \bar{\varsigma})$ (i.e., for $n = 10$) and select events with $\Lambda^\varsigma$ above a certain threshold. The conditional probability $P(\sigma' = \varsigma | \varsigma = \downarrow (\uparrow))$ is then estimated following the procedure described above (but with more ancilla measurements, see Methods). Indeed, $F_P^\downarrow$ increases from 94 to 99% at the median (for $\Lambda^\downarrow > 1$), and $F_P^\downarrow$ reaches 99.6% at the 76th percentile, see Fig. 4. The limiting value is higher for the spin-down case, as expected from $F_{QND}^{\downarrow(\uparrow)}$.

## Discussion

In the present experiment, 30 ancilla measurements are feasible before we lose strong correlation between the input and the outcome ($F_{QND}^\uparrow \lesssim 50\%$). This is limited by a relatively short electron spin lifetime, compared with single nuclear spins in silicon where 99.8% readout fidelity is achieved as a result of >99.98% non-demolition fidelity[19,20]. We note that, while both $F_M$ and $F_P$ are successfully improved by the cumulative QND readout, the observed $F_{QND}$ falls short of our earlier expectations[18] and the overall QND performance is impacted by this. The ratio $T_1^\downarrow / T_1^\uparrow = 31$ is deviated from the ideal thermal population ratio (=16) between the Zeeman sublevels at the electron temperature (~50 mK), and the measured $T_1^\uparrow$ is roughly 30 times shorter than nominal expectation for an idle spin away from the hotspot[21]. Indeed, data imply that the qubit relaxation occurs predominantly during the ancilla readout process (see Supplementary Fig. 1). This effect is expected to be suppressed by further quenching the residual exchange coupling (~MHz), e.g., via an interdot gate electrode[6] or by fast readout with an ancilla encoded in double-dot spin states[22]. We anticipate that we will then improve $F_{QND}$ and the QND readout in all aspects, as a higher $F_{QND}$ should raise $F_M$ and $F_P$ that are achievable by repeating QND measurements.

$F_M$ and $F_P$ will also improve, particularly for small numbers of $n$, by decreasing single-shot QND measurement infidelities $1 - f_i^{\downarrow(\uparrow)}$, which are 15% (25%) on average for $i = 1–20$. We estimate the contribution of charge discrimination error to be 1% (7%) for the spin-down (up) case (see Supplementary Fig. 3), which can be straightforwardly reduced by tuning the charge

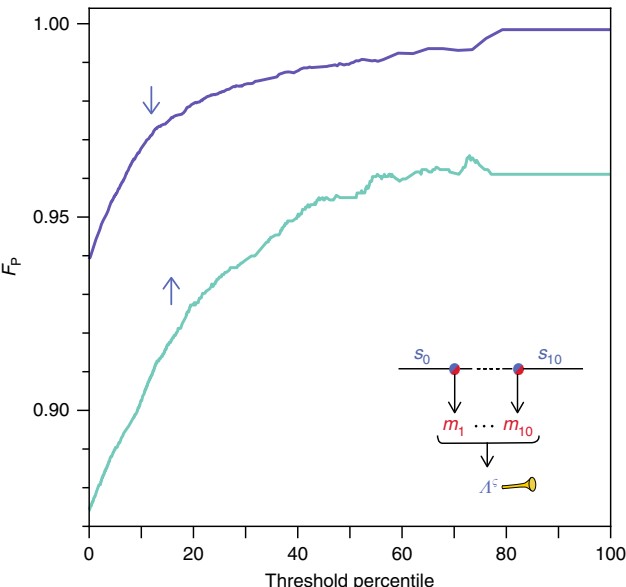

**Fig. 4 Heralded enhanced preparation fidelity.** Events with high initialization confidence are selected based on $\Lambda^\varsigma$. The data include 53,000 events in total and threshold percentiles with more than 5000 selected events are used for the analysis.

sensor sensitivity solely for the ancilla dot. The remaining single-shot infidelities (in converting the qubit spin state to an ancilla electron tunneling event) are more significant, 14% and 18% for the spin-down and -up cases, respectively. We believe that these arise from the qubit-ancilla conditional operation and the ancilla spin-to-charge conversion and initialization process, and can be addressed by optimizing the two-qubit gate operation and the spin-selective tunneling process[4,10].

To conclude, we have demonstrated a QND readout of a single electron spin in silicon. The presented technique uses an electron spin in a neighboring dot as an ancilla, requiring no increased structural complexity to multiple-dot quantum information processing units. Central to the 99% non-demolition fidelity are a synthesized Ising type qubit-ancilla coupling and the rapid conditional ancilla rotation and measurement. The ancilla-based QND readout is a crucial element in qubit error detection and correction protocols. More specifically, it should be naturally extensible to QND measurements of the parity of multiple qubits with proper choice of single- and two-qubit gate operations, in contrast to the spin blockade-based single-shot readout of a single spin[23], which may also allow for repetitive readout[24]. Combined with high-fidelity single- and two-qubit gates[2,4], the demonstrated results will pave the way toward fault-tolerant quantum-information processing in the silicon quantum-dot platform.

## Methods

**Measurement setup.** The device is a dual-gated accumulation-mode Si/SiGe quantum dot reported in ref. [11] and is measured in a dry dilution refrigerator (Oxford Instruments Triton 200). A Tektronix AWG5014 arbitrary waveform generator is used to generate three-channel gate pulses (applied to $V_C$, $V_T$, and $V_B$). To ensure the adiabaticity of the pulses, they are filtered through Bessel analog filters with a 3 dB cutoff frequency at 39 MHz. The AWG5014 triggers a Tabor WX2184 waveform generator which produces I/Q modulation waveforms for two Keysight 8267D microwave sources. We use single-sideband modulation at 20 MHz to suppress the effects from leakage and spurious modes. In order to maintain the device in the symmetric condition throughout a controlled-rotation operation, the pulse heights for $V_C$, $V_T$, and $V_B$ are chosen to be +30.0 mV, −23.1 and −21.0 mV, respectively[15]. Each spin is read out using the energy-selective tunneling to the adjacent reservoir. (The ancilla and qubit electrons tunnel in and out from different reservoirs.) The reflectometry signal (at 205 MHz) is demodulated to baseband, sampled by an AlazarTech digitizer ATS9440 at 10 MSPS,

filtered at 1 MHz using a second order Butterworth digital filter and decimated at 2 MSPS for post processing. The lengths of individual traces are 45 μs for experiments in Fig. 1c and 30 μs for those in Fig. 2. Peak-to-peak values (the difference between the maximum and the minimum readings in individual traces) are used to detect the tunneling events.

**Bayesian models.** We construct a cumulative QND readout from $n$ consecutive outcomes ($\boldsymbol{m}_n$) of ancilla measurements (Fig. 3) based on the performance of single-shot QND measurements ($m_i$) characterized using Eq. (1) as described in the main text. In order to analyze all joint probabilities in a consistent manner, the fitting is performed in the following steps. First, the joint probability $P(m_i m_{30})$ for each value of $i$ is fit to Eq. (1) with $p_{i-1}^\downarrow (= 1 - p_{i-1}^\uparrow)$, $f_i^{\downarrow(\uparrow)}$, and $g_i^{\downarrow(\uparrow)}$ as fitting parameters, assuming that $p_{i-1}^\downarrow$ is an exponentially decaying Rabi oscillation[11], similarly to ref. [18]. We then model the $i$-dependence of $p_i^\downarrow$ as $p_{i+1}^\downarrow = \rho^\downarrow p_i^\downarrow + (1 - \rho^\uparrow) p_i^\downarrow$, where $\rho^{\downarrow(\uparrow)} = \exp\left(-60\,\mu\text{s}/T_1^{\downarrow(\uparrow)}\right)$ is the spin conservation probability and $p_0^\downarrow$ is parametrized using an exponentially decaying Rabi oscillation again. Finally, we fix $p_{i-1}^\downarrow$ to the values calculated from $p_0^\downarrow$ and $\rho^{\downarrow(\uparrow)}$ in the model above and fit $P(m_i m_{30})$ for each value of $i$ with only $f_i^{\downarrow(\uparrow)}$ and $g_i^{\downarrow(\uparrow)}$ as fitting parameters. Values extracted in these initial and later analysis steps are compared in Supplementary Fig. 2.

When we regard a cumulative QND readout process as a measurement device, it should estimate from $\boldsymbol{m}_n$ the input spin state $s_0$ to be either ↓ or ↑. Our goal is to precisely determine the estimator $\sigma$ for $s_0$ such that $s_0$ is more likely $\sigma$ given $\boldsymbol{m}_n$, i.e., $P(s_0 = \sigma | \boldsymbol{m}_n) > P(s_0 = \bar{\sigma} | \boldsymbol{m}_n)$, assuming no prior knowledge about the probability distribution of $P(s_0)$, i.e., $P(s_0 = \downarrow) = P(s_0 = \uparrow) = 1/2$. From the Bayes theorem, we see that for the more likely value of $\sigma$

$$P(\boldsymbol{m}_n | s_0 = \sigma) = \frac{P(s_0 = \sigma | \boldsymbol{m}_n) P(\boldsymbol{m}_n)}{P(s_0 = \sigma)} > \frac{P(s_0 = \bar{\sigma} | \boldsymbol{m}_n) P(\boldsymbol{m}_n)}{P(s_0 = \bar{\sigma})} = P(\boldsymbol{m}_n | s_0 = \bar{\sigma}), \tag{2}$$

meaning that the input state estimation comes down to comparing likelihoods $P(\boldsymbol{m}_n | s_0 = \downarrow)$ and $P(\boldsymbol{m}_n | s_0 = \uparrow)$. For optimal performance, we should consider all ($2^{n-1}$) possible spin trajectories $\{s_1, s_2, \dots s_{n-1}\}$ following $s_0$, with the realization probabilities taken into account. Using the spin transition probability to calculate the realization probabilities, the likelihood $P(\boldsymbol{m}_n | s_0)$ can be computed as

$$P(\boldsymbol{m}_1 | s_0) = \Theta_{s_0, m_1}\left(f_1^{s_0}\right) \tag{3}$$

for $n = 1$ and for $n > 1$

$$P(\boldsymbol{m}_n | s_0) = \sum_{s_1 = \downarrow}^{\uparrow} \cdots \sum_{s_{n-1} = \downarrow}^{\uparrow} \Theta_{s_{n-1}, m_n}\left(f_n^{s_{n-1}}\right) \prod_{i=1}^{n-1} \Theta_{s_{i-1}, s_i}(\rho^{s_{i-1}}) \Theta_{s_{i-1}, m_i}\left(f_i^{s_{i-1}}\right). \tag{4}$$

When we instead view a cumulative QND readout process as a state preparation device, it prepares the posterior spin state $s_n$ in a specific state, either ↓ or ↑, as determined from $\boldsymbol{m}_n$. Our task is then to construct the estimator $\varsigma$ for $s_n$ such that $s_n$ is more likely $\varsigma$ given $\boldsymbol{m}_n$, i.e., $P(s_n = \varsigma | \boldsymbol{m}_n) > P(s_n = \bar{\varsigma} | \boldsymbol{m}_n)$, assuming no prior knowledge about the probability distribution of $P(s_n)$, i.e., $P(s_n = \downarrow) = P(s_n = \uparrow) = 1/2$. From the Bayes theorem, it follows for the more likely value of $\varsigma$

$$P(\boldsymbol{m}_n | s_n = \varsigma) = \frac{P(s_n = \varsigma | \boldsymbol{m}_n) P(\boldsymbol{m}_n)}{P(s_n = \varsigma)} > \frac{P(s_n = \bar{\varsigma} | \boldsymbol{m}_n) P(\boldsymbol{m}_n)}{P(s_n = \bar{\varsigma})} = P(\boldsymbol{m}_n | s_n = \bar{\varsigma}). \tag{5}$$

This means that we should now compare the likelihoods $P(\boldsymbol{m}_n | s_n = \downarrow)$ and $P(\boldsymbol{m}_n | s_n = \uparrow)$. Again, for optimal performance, we need to sum up probabilities of all ($2^n$) possible spin trajectories $\{s_0, s_1, \dots s_{n-1}\}$ ending with $s_n = \varsigma$ as

$$P(\boldsymbol{m}_n | s_n) = \sum_{s_0 = \downarrow}^{\uparrow} \cdots \sum_{s_{n-1} = \downarrow}^{\uparrow} \prod_{i=1}^{n} \Theta_{s_{i-1}, s_i}(\rho^{s_{i-1}}) \Theta_{s_{i-1}, m_i}\left(f_i^{s_{i-1}}\right). \tag{6}$$

In the main text, these Bayesian models are used to determine $\sigma$ and $\varsigma$.

**Conversion between conditional probabilities.** As explained in the main text, $F_P^{\downarrow(\uparrow)}$ is defined by the conditional probability $P(s_n = \varsigma | \varsigma = \downarrow (\uparrow))$. Experimentally, $s_n$ can only be estimated with a finite fidelity from the ancilla measurement results $\boldsymbol{m}'_{j,k} = \left\{ m_j, m_{j+1}, \cdots, m_{j+k-1} \right\}$ with $j \geq n + 1$. We denote such an estimator for $s_n$ as $\sigma'$ and its measurement fidelity as $F_M^{\prime s_n} \equiv P(\sigma' = s_n | s_n)$. We can then convert $P(\sigma' = \varsigma | \varsigma)$ and $F_M^{\prime s_n}$ to $F_P^\varsigma$, by noting that for $\varsigma = \downarrow$ and ↑,

$$\begin{aligned} P(\sigma' = \varsigma | \varsigma) &= P(\sigma' = \varsigma | s_n = \varsigma) P(s_n = \varsigma | \varsigma) + P(\sigma' = \varsigma | s_n = \bar{\varsigma}) P(s_n = \bar{\varsigma} | \varsigma) \\ &= F_M^{\prime \varsigma} F_P^\varsigma + (1 - F_M^{\prime \varsigma})(1 - F_P^\varsigma). \end{aligned} \tag{7}$$

Larger $j - n$ and $k$ would make the result more robust against correlated measurement errors and statistical shot noise, but $j + k$ cannot exceed 30 (in the current experiment). We employ $(j, k) = (n + 4, 5)$ for Fig. 3e so that we can measure up to $n = 20$, and $(n + 4, 15)$ for Fig. 4 in order to obtain statistically more reliable results for high percentiles.

## Data availability

The data that support the findings of this study are available from the corresponding authors upon reasonable request.

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

## Acknowledgements

We thank Microwave Research Group at Caltech for technical assistance. Part of this work was financially supported by CREST, JST (JPMJCR15N2, JPMJCR1675), the ImPACT Program of Council for Science, Technology and Innovation (Cabinet Office, Government of Japan), MEXT Quantum Leap Flagship Program (MEXT Q-LEAP) Grant Number JPMXS0118069228, JSPS KAKENHI Grants Nos. 26220710, 17K14078, 18H01819, and 19K14640, RIKEN Incentive Research Projects and The Murata Science Foundation.

## Author contributions

J.Y. acquired and analyzed the data and wrote the paper. K.T. and A.N. set up the measurement hardware with assistance from J.Y., T.N., and S.L. T.N. contributed to the data analysis. K.T. fabricated the device with help from J.K. and T.K. S.T. supervised the project.

## Competing interests

The authors declare no competing interests.
