## [Peer Review File · Nature Communications]

Reviewers' comments:

Reviewer #1 (Remarks to the Author):

In this manuscript, the authors demonstrate quantum non-demolition (QND) readout in a Si/SiGe double dot device with natural ^{29}Si composition and a superimposed micromagnet that establishes a magnetic field gradient suitable for realizing electric dipole spin resonance. Using one of these two electron spins as an ancilla qubit that may be coupled to the primary (single electron spin) qubit of interest, the authors perform repeated two-qubit (CNOT) gates followed by ancilla measurement to realize the QND measurement.

While a similar QND technique was demonstrated by a subset of the authors and collaborators in a recent publication (Ref. [17]), that was implemented in a GaAs/AlGaAs quantum well rather than silicon. As far as I am aware, this is the first demonstration of QND of electron spins in silicon, a material that poses some practical challenges that are either not present or reduced in GaAs. Considering the importance of QND measurements to the maturation of quantum information processing (QIP) in silicon, the demonstration of this technique in silicon should be of interest to the community.

The authors have performed a careful Bayesian statistical analysis of the three error probabilities of primary interest in the QND measurement context, namely (1) non-demolition, (2) measurement, and (3) preparation. The statistical analysis appears to follow closely that of Ref [17], with the addition of implementation and discussion of heralded enhancement of preparation fidelity. I find that this analysis is clear and convincing, illustrating some of the fundamental limiters to the measurement fidelity such as a short spin relaxation time.

Overall this work demonstrates an important step forward in maturing QIP in silicon and should be of interest to the broader community.

Reviewer #2 (Remarks to the Author):

This manuscript revolves around the experimental realisation of quantum non-demolition (QND) measurements applied to an electron spin qubit in silicon. Such QND measurements are critical to many key quantum information protocols, such as error correction, and could enable large-scale quantum processors. Hence, I find this work of importance for the field. It follows the recent demonstration of a similar QND protocol in GaAs spin qubits [Ref 17] from some of the authors. Strictly speaking, this is not the first realisation of QND in silicon either, given that this type of measurement was demonstrated in 2013 for a donor-based nuclear spin qubit [Ref 18]. However, the realisation of QND in quantum dots shown here is inherently more scalable and may allow more realistic technological advances. However, in the present form, I do not recommend publication because there are a number of unanswered issues, which I summarise next.

1) While the experiments are expertly carried out and the demonstration of QND is convincing, I am concerned that the authors failed to show an increase of the overall fidelity by repeating the protocol. Actually, as far as I can see in Fig. 3b, they show a decrease of average fidelity as a function of the number of measurements. This is at odds with what the authors showed in Ref 17. I feel that the lack of this QND fidelity boost (one of the most attractive features of this protocol) is not sufficiently justified and explained. I remain uncertain of the causes. It seems that the authors blame short spin lifetime for this. But this just shifts the question towards: why there was such an unexpectedly short spin lifetime in this experiment? If, for instance, the ancilla readout results in an enhancement of the qubit relaxation rate, it is not clear to me that this QND protocol would ever be of any practical use. The authors need to make a convincing case that there is nothing inherently detrimental in the used technique.

2) Another issue to be resolved is the lack of a compare-and-contrast discussion between this

work and the much-related Ref. 17 published earlier this year from the same authors. They have been in the privileged position of pioneering the QND protocol in the near-ideal test bed provided by GaAs. It would be very informative for the community to understand what specific challenges have been encountered in its translation onto Si. In Ref. 17, the authors produced simulations of how the repetitive nature of the protocol would have been even more beneficial in Si than in GaAs. It would be valuable to elaborate on this aspect in this manuscript, particularly because those early simulations do not seem to have been a good oracle in light of the results reported here where a QND fidelity boost was not shown.

3) I would like to see an outlook for this technique's future use in scaled-up multi-qubit devices. The authors briefly mention that QND measurements in silicon quantum dots could be improved by adding extra gates to fine tune J , or using a S-T ancilla as opposed to the single-spin ancilla used here. Hence, a number of questions arise: A) how a two-qubit cell should look like in a silicon quantum processor that relies on QND and error correction? B) the lack of QND boost was inherent to the device architecture that was not carefully optimised and how this can be fixed? C) the problem with the reduced sensor's sensitivity (see also point 9 below) is solvable at the expense of a more cumbersome measurement protocol or other type of sensors would work better (e.g. gate-base rf readout)? A concluding paragraph touching upon some of these unanswered questions would help.

Some additional minor improvements are also to be considered:

4) I think the title should contain info about the QND nature of the repetitive measurements. I find it ambiguously worded at the moment.

5) Some more details about the spin-to-charge conversion protocol used for readout are needed. Is spin-dependent tunnelling occurring between the dots and the reservoirs or the dot and the sensor? Do the ancilla and the qubit electrons tunnel in/out from different reservoirs? A citation to Elzerman's Nature paper where the spin-dependent tunnelling was pioneered is missing.

6) What dictated the choice of $t_{CZ}=0.53$ us? Can the authors justify this and show this operating point in Fig. 1d?

7) It would be informative to show the position of the micromagnet in Fig. 1a.

8) In Fig. 2c and Fig. 3b-e, it is very difficult to discriminate the different traces because colors and data symbols are too similar.

9) In the caption of Fig. 2, the authors justify a reduced visibility of the Rabi oscillations as the protocol progresses with a loss of sensitivity of the charge detector. Firstly, this is important information that should not be relegated to the caption, but plainly discussed and explained in the main text. Secondly, why a conventional feedbacked compensation pulse was not applied to the detector's gate to keep it at the point of maximum sensitivity at all times?

10) In Fig 3c, it seems that the oscillation's visibility goes up as a function of i . It would be useful to plot visibility vs i .

Reviewer #3 (Remarks to the Author):

This paper demonstrates quantum non-demolition readout of a single electron spin in silicon. The authors are adapting a technique they demonstrated in Ref. 17 to perform a qubit-state-dependent rotation on an ancilla qubit. By measuring the phase accrued on the ancilla, they obtain information about the qubit. This process does not significantly disturb the qubit beyond projection, so they are able to repeat it many times to improve the measurement fidelity. They also use this process to demonstrate high-fidelity heralded state preparation. This is high-quality work, and I recommend publication in Nature Communications. It adds another nice capability to the silicon spin-qubit arsenal. In my view, the only potential argument against publication in Nature Communications is that this work is similar in spirit to Ref. [17], but that reference used a GaAs spin qubit and a different type of ancilla qubit.

My main question for the authors is the following: How does this readout method practically compare to the case of reading out a single spin via a spin-blockade measurement with a classical ancilla spin in a magnetic gradient? In a magnetic gradient, the two-spin combinations UU and DU (the first is the qubit and the second is the ancilla) can be mapped to different charge configurations via spin blockade and read out with high fidelity. This idea has been mentioned in the literature, and I think the research group of the authors has used this before. I think this is a non-demolition readout method, and in principle it can be read out very quickly and with very high fidelity, even in silicon, as the reflectometry paper from the research group of the authors has recently demonstrated.

My second question has to do with readout fidelity. It seems like the authors are assuming that the qubit evolution should follow an exponentially damped Rabi oscillation, and that any errors result from state preparation or readout errors. I believe that these estimated readout errors have been used later in the analysis of the non-demolition readout. Has it indeed been confirmed that the qubit should follow an exponentially damped Rabi oscillation? Are the only errors preparation and readout? For example, what is the magnitude of manipulation errors? Or leakage errors? Or rotating-frame relaxation?

Last, what is the fidelity of the CZ gate, and how does it affect the overall non-demolition metrics?

I imagine that the errors associated with these effects are small, but this paper reports very high fidelities, and these small sources of error may be worth considering. In general, it would be nice to see an explicit listing of how much error comes from what mechanism.

We thank the Reviewers for carefully reading our manuscript and for helpful comments and suggestions. In this response letter, we address the Reviewers' concerns in a point-by-point manner. All changes in the manuscript (except those of the reference numbers) are highlighted in red. We also note that Figures 1a, 1d, and S2 are modified and colours in Figures 2c, 3b, 3d, and 3e are changed.

Reply to Reviewer #1

In this manuscript, the authors demonstrate quantum non-demolition (QND) readout in a Si/SiGe double dot device with natural ^{29}Si composition and a superimposed micromagnet that establishes a magnetic field gradient suitable for realizing electric dipole spin resonance. Using one of these two electron spins as an ancilla qubit that may be coupled to the primary (single electron spin) qubit of interest, the authors perform repeated two-qubit (CNOT) gates followed by ancilla measurement to realize the QND measurement.

While a similar QND technique was demonstrated by a subset of the authors and collaborators in a recent publication (Ref. [17]), that was implemented in a GaAs/AlGaAs quantum well rather than silicon. As far as I am aware, this is the first demonstration of QND of electron spins in silicon, a material that poses some practical challenges that are either not present or reduced in GaAs. Considering the importance of QND measurements to the maturation of quantum information processing (QIP) in silicon, the demonstration of this technique in silicon should be of interest to the community.

The authors have performed a careful Bayesian statistical analysis of the three error probabilities of primary interest in the QND measurement context, namely (1) non-demolition, (2) measurement, and (3) preparation. The statistical analysis appears to follow closely that of Ref [17], with the addition of implementation and discussion of heralded enhancement of preparation fidelity. I find that this analysis is clear and convincing, illustrating some of the fundamental limiters to the measurement fidelity such as a short spin relaxation time.

Overall this work demonstrates an important step forward in maturing QIP in silicon and should be of interest to the broader community.

We really appreciate the Reviewer's positive and knowledgeable assessment of our work.

Reply to Reviewer #2

This manuscript revolves around the experimental realisation of quantum non-demolition (QND) measurements applied to an electron spin qubit in silicon. Such QND measurements are critical to many key quantum information protocols, such as error correction, and could enable large-scale quantum processors. Hence, I find this work of importance for the field. It follows the recent demonstration of a similar QND protocol in GaAs spin qubits [Ref 17] from some of the authors. Strictly speaking, this is not the first realisation of QND in silicon either, given that this type of measurement was demonstrated in 2013 for a donor-based nuclear spin qubit [Ref

18]. However, the realisation of QND in quantum dots shown here is inherently more scalable and may allow more realistic technological advances. However, in the present form, I do not recommend publication because there are a number of unanswered issues, which I summarise next.

We thank the Reviewer for appreciating the impact of our first demonstration of QND readout of an electron spin in silicon.

1) While the experiments are expertly carried out and the demonstration of QND is convincing, I am concerned that the authors failed to show an increase of the overall fidelity by repeating the protocol. Actually, as far as I can see in Fig. 3b, they show a decrease of average fidelity as a function of the number of measurements. This is at odds with what the authors showed in Ref 17. I feel that the lack of this QND fidelity boost (one of the most attractive features of this protocol) is not sufficiently justified and explained. I remain uncertain of the causes. It seems that the authors blame short spin lifetime for this. But this just shifts the question towards: why there was such an unexpectedly short spin lifetime in this experiment? If, for instance, the ancilla readout results in an enhancement of the qubit relaxation rate, it is not clear to me that this QND protocol would ever be of any practical use. The authors need to make a convincing case that there is nothing inherently detrimental in the used technique.

We understand the Reviewer's concern. As stated in the introductory paragraph, one of the main attractive features of QND measurements is the fidelity boost, but this should only be expected for the measurement and preparation fidelities. We indeed demonstrated this in Figs. 3c, d and e, which proves that our protocol is of practical use. (Note that the increase of the overall measurement fidelity shown in Fig. 3d corresponds to what was shown in Fig. 3c of Ref. 17 (18 in the revised manuscript).) The non-demolition fidelity F_{QND} , in contrast, *should not* increase (but decrease monotonically or remain the same at best) as we increase the number of measurements (denoted by n in the main text). This follows from the definition – it quantifies the degree to which the initial (projected) qubit state is preserved. We agree that this was not clearly stated in the previous manuscript. We therefore added the following phrase to the last paragraph on page 3 that explains Fig. 3b: “unlike the other two fidelities, it is expected to decrease as n is increased”.

2) Another issue to be resolved is the lack of a compare-and-contrast discussion between this work and the much-related Ref. 17 published earlier this year from the same authors. They have been in the privileged position of pioneering the QND protocol in the near-ideal test bed provided by GaAs. It would be very informative for the community to understand what specific challenges have been encountered in its translation onto Si. In Ref. 17, the authors produced simulations of how the repetitive nature of the protocol would have been even more beneficial in Si than in GaAs. It would be valuable to elaborate on this aspect in this manuscript, particularly because those early simulations do not seem to have been a good oracle in light of the results reported here where a QND fidelity boost was not shown.

We thank the referee for the constructive comments. It would be beneficial to contrast the difference between the demonstrated result and the simulated performance based on the GaAs test bed. As we explained in our response to 1), we *did* show the measurement/preparation fidelity boost by cumulative QND readout. Furthermore, the benefit of measurement repetition is indeed enhanced for the Si device, consistent with the prediction we made in Ref. 17 (18 in the revised manuscript). We note that our previous study assumed a different ancilla readout technique and one of the best single-shot readout fidelities in the literature, which should guide towards improved cumulative QND measurement fidelities in future experiments. On the other hand, what we did not expect prior to the experiment is the increased spin relaxation during ancilla measurements, and in the main text we discussed how we would be able to suppress it in next devices (the first paragraph of page 5). To highlight this, in the revised manuscript, we added the following sentence to the last paragraph of page 4 where the relevant topic is discussed: “We note that, while both F_M and F_P are successfully improved by the cumulative QND readout, the observed F_{QND} falls short of our earlier expectations [18] and the overall QND performance is impacted by this.”

3) I would like to see an outlook for this technique’s future use in scaled-up multi-qubit devices. The authors briefly mention that QND measurements in silicon quantum dots could be improved by adding extra gates to fine tune J , or using a S-T ancilla as opposed to the single-spin ancilla used here. Hence, a number of questions arise: A) how a two-qubit cell should look like in a silicon quantum processor that relies on QND and error correction? B) the lack of QND boost was inherent to the device architecture that was not carefully optimised and how this can be fixed? C) the problem with the reduced sensor’s sensitivity (see also point 9 below) is solvable at the expense of a more cumbersome measurement protocol or other type of sensors would work better (e.g. gate-base rf readout)? A concluding paragraph touching upon some of these unanswered questions would help.

We thank the Reviewer for the helpful suggestions. We agree that some outlook of this technique in multi-qubit devices would be informative. Regarding A), one of the advantages of the presented scheme is that it does not involve a different type of qubit as an ancilla, so that it will not constrain the “unit” structure design. We agree that this should be touched upon in the main manuscript. As for B), we *did* demonstrate the QND boost of the measurement and preparation fidelities as expected (see also our response to 1) above). We believe that the key ingredients would be the Ising-type coupling and the rapid ancilla measurement. With regard to C), in the present experiment the sensor sensitivity was slightly compromised so as to also perform the destructive qubit readout (see also our response to 9) below). This was only necessary for demonstration purposes to prove the consistency of the ancilla-based readout with the conventional technique. We believe that the effect of sensor signal-to-noise ratio (charge discrimination error) is discussed adequately in the revised manuscript (please see also our response to the last point raised by Reviewer #3). We have added the following sentences to the revised concluding paragraph on page 5: “To conclude, we have for the first time

demonstrated a QND readout of a single electron spin in Si. The presented technique uses an electron spin in a neighbouring dot as an ancilla, requiring no increased structural complexity to multiple-dot quantum information processing units. Central to the 99% non-demolition fidelity are a synthesized Ising type qubit-ancilla coupling and the rapid conditional ancilla rotation and measurement.”

Some additional minor improvements are also to be considered:

4) I think the title should contain info about the QND nature of the repetitive measurements. I find it ambiguously worded at the moment.

We understand the Reviewer’s concern and have changed the title to “Quantum non-demolition readout of an electron spin in silicon”.

5) Some more details about the spin-to-charge conversion protocol used for readout are needed. Is spin-dependent tunnelling occurring between the dots and the reservoirs or the dot and the sensor? Do the ancilla and the qubit electrons tunnel in/out from different reservoirs? A citation to Elzerman’s Nature paper where the spin-dependent tunnelling was pioneered is missing.

We agree with the Reviewer that the spin-to-charge conversion process may not have been explained sufficiently in detail. We added the following sentence to Method which is referred to in the third paragraph of page 2: “The ancilla and qubit electrons tunnel in and out from different reservoirs.” We also added a citation to Elzerman’s Nature paper as suggested by the Reviewer.

6) What dictated the choice of $t_{CZ}=0.53$ us? Can the authors justify this and show this operating point in Fig. 1d?

We find this a helpful suggestion. The choice of $t_{CZ}=0.53$ us simply follows from $1/(2J)$, which we now explicitly point out in the main text. We have added a vertical line to Fig. 1d to visualize that $t_{CZ} = 0.53$ us corresponds to a phase shift of π (which results in a conditional π rotation as explained in the main text).

7) It would be informative to show the position of the micromagnet in Fig. 1a.

Following this suggestion, we now display the shape of the micromagnet in Fig. 1a.

8) In Fig. 2c and Fig. 3b-e, it is very difficult to discriminate the different traces because colors and data symbols are too similar.

We thank the Reviewer for this suggestion. We have made the colours more distinct from each other in these Figures.

9) In the caption of Fig. 2, the authors justify a reduced visibility of the Rabi oscillations as the protocol progresses with a loss of sensitivity of the charge detector. Firstly, this is important information that should not be relegated to the caption, but plainly discussed and explained in the main text. Secondly, why a conventional feedbacked compensation pulse was not applied to the detector's gate to keep it at the point of maximum sensitivity at all times?

We apologize for a possible confusion caused by the lack of detailed protocol for the destructive readout of the qubit as pointed out by the Reviewer in 5). As now mentioned in Methods of the revised manuscript, the destructive qubit readout (labelled as q in Fig. 2) is performed using the spin-selective tunnelling from the qubit dot to its closest reservoir. Such tunnelling events naturally generate smaller sensor signals given the larger distance from the sensor. This is the cause of the reduced sensitivity for q as compared to m_{30} . We briefly mention this in the figure caption because it helps to better understand the figure, while the reduced visibility of q itself is inessential to the main content. After all, q is only measured to confirm that our ancilla measurement results (m_i) are consistent with the conventional readout. We note that as mentioned in the main text (the second paragraph of page 3) the shot-by-shot fidelity of ancilla measurements (f_i) plotted in Extended Data Figure 1b are essentially i -independent (i.e. constant as the repeated QND readout protocol progresses), hence the sensor sensitivity. The reduction of Rabi oscillation visibility as the QND readout process progresses is fully accounted for by the non-demolition fidelity (F_{QND}) and no sensor sensitivity reduction is assumed in our analysis.

10) In Fig 3c, it seems that the oscillation's visibility goes up as a function of i . It would be useful to plot visibility vs i .

We find this a nice suggestion to consider. However, our conclusion is that it would also most likely mislead the reader, since the oscillation visibility can be easily mistaken as the measurement visibility. The latter is far more relevant here and is essentially the average measurement fidelity plotted in Fig. 3d, which we now mention in the caption of Fig. 3d.

Reply to Reviewer #3

This paper demonstrates quantum non-demolition readout of a single electron spin in silicon. The authors are adapting a technique they demonstrated in Ref. 17 to perform a qubit-state-dependent rotation on an ancilla qubit. By measuring the phase accrued on the ancilla, they obtain information about the qubit. This process does not significantly disturb the qubit beyond projection, so they are able to repeat it many times to improve the measurement fidelity. They also use this process to demonstrate high-fidelity heralded state preparation. This is high-quality work, and I recommend publication in Nature Communications. It adds another nice capability to the silicon spin-qubit arsenal. In my view, the only potential argument against publication in Nature Communications is that this work is similar in spirit to Ref. [17], but that reference used a GaAs spin qubit and a different type of ancilla qubit.

We thank the Reviewer for their careful and thorough review as well as for recommending publication in Nature Communications.

My main question for the authors is the following: How does this readout method practically compare to the case of reading out a single spin via a spin-blockade measurement with a classical ancilla spin in a magnetic gradient? In a magnetic gradient, the two-spin combinations UU and DU (the first is the qubit and the second is the ancilla) can be mapped to different charge configurations via spin blockade and read out with high fidelity. This idea has been mentioned in the literature, and I think the research group of the authors has used this before. I think this is a non-demolition readout method, and in principle it can be read out very quickly and with very high fidelity, even in silicon, as the reflectometry paper from the research group of the authors has recently demonstrated.

We appreciate the constructive and quite relevant comment. As the Reviewer pointed out, an alternative way to readout a single spin is to use a spin-blockade measurement with the spin in the neighbouring dot as a reference and we have routinely used this technique. This can be made non-destructive in the sense that the electron will not be lost, but strictly speaking, it will not possess the QND character as the measured spin observable is not preserved throughout the entire readout process. Importantly, in contrast to the presented scheme, it would *not* be naturally applicable to QND measurements of multi-qubit parity required in many key quantum information protocols including error correction. However, it might instead allow a repetitive measurement of a single spin. We agree with the Reviewer that this is a highly relevant point so we have added the following sentence to the concluding paragraph in the main text (page 5): “More specifically, it should be naturally extensible to QND measurements of the parity of multiple qubits with proper choice of single- and two-qubit gate operations, in contrast to the spin blockade-based single-shot readout of a single spin [23], which may also allow for repetitive readout [24].”

My second question has to do with readout fidelity. It seems like the authors are assuming that the qubit evolution should follow an exponentially damped Rabi oscillation, and that any errors result from state preparation or readout errors. I believe that these estimated readout errors have been used later in the analysis of the non-demolition readout. Has it indeed been confirmed that the qubit should follow an exponentially damped Rabi oscillation? Are the only errors preparation and readout? For example, what is the magnitude of manipulation errors? Or leakage errors? Or rotating-frame relaxation?

We thank the Reviewer for pointing this out. The qubit spin is confirmed to follow an exponentially damped Rabi oscillation from the conventional destructive readout. This has been observed and more closely investigated in our early study (Ref. 11) in the same system, which we now also cite where this is discussed (the second paragraph of page 3). The probability distribution of the prepared qubit spin state, $p_{i-1}^{\downarrow(1)}$ includes all kinds of errors in the qubit spin state preparation, such as manipulation errors, rotating-frame relaxation (Rabi decay) and spin initialization errors. The joint

probability analysis (e.g. Eq (1)) is used to separate these from the measurement error. We realized that in the original manuscript what we meant by “the preparation error” in this context was ambiguous so this part is rephased as follows: “We separate the error in the prepared qubit spin state (during the process of initialization, rotation and preceding ancilla measurements) from the measurement infidelity [18] by expressing the joint probability as ...”

Last, what is the fidelity of the CZ gate, and how does it affect the overall non-demolition metrics?

I imagine that the errors associated with these effects are small, but this paper reports very high fidelities, and these small sources of error may be worth considering. In general, it would be nice to see an explicit listing of how much error comes from what mechanism.

We appreciate the Reviewer’s constructive suggestion. In Supplementary Note 2, we separate the charge discrimination error (in the ancilla charge detection process) from the rest, i.e. the conversion error from the qubit spin state into an ancilla electron tunnelling event. The result was originally mentioned only briefly, so we now explicitly explain that the contribution from the qubit-to-ancilla-blip conversion errors is considerably larger than the (ancilla) charge discrimination error. To help visualize this, we have added the plots of sensor signal distributions for given qubit spin states (Fig. S2d) and added some explanations for this to the Note.

The qubit-to-ancilla-blip conversion errors could arise from the qubit-ancilla entangling conditional operation, the ancilla-spin initialization process, and the ancilla spin-to-charge conversion. However, we did not perform any elaborate characterization of the classical fidelity of the conditional rotation that is designed to be free from the SPAM errors. The other two mechanisms are also intricate to quantify separately. While it is of practical interest to improve these, the cumulative QND readout performance is at present impacted by the (unexpected) enhanced spin relaxation during ancilla measurements and hence F_{QND} . Therefore, while agreeing with the Reviewer that it would be nice to be able to break down individual error contributions, we argue that it is a scope of future study.

As a result, we now cite the explicit values of overall qubit-to-ancilla-blip conversion errors, state that they are more significant than the discrimination errors, and list the possible causes in the main text (the second paragraph of page 5). We also added the following phrase (to the last paragraph of page 4) to emphasize the significance of F_{QND} to the QND performance: “(single nuclear spins in silicon where 99.8% readout fidelity is achieved) as a result of > 99.98% non-demolition fidelity. We note that the observed F_{QND} ... the overall QND performance is impacted by this.”

REVIEWERS' COMMENTS:

Reviewer #2 (Remarks to the Author):

I am satisfied with the author's modifications and recommend publication as is.

Reviewer #3 (Remarks to the Author):

I believe that the authors have addressed all of the points raised by the reviewers satisfactorily. I am happy to recommend publication in Nature Communications.